# A lower psoas muscle volume was associated with a higher rate of recurrence in male clear cell renal cell carcinoma

Go Noguchi[1], Takashi Kawahara[2]*, Kota Kobayashi[1], Sohgo Tsutsumi[1], Shinji Ohtake[3], Kimito Osaka[1], Susumu Umemoto[1], Noboru Nakaigawa[3], Hiroji Uemura[2], Takeshi Kishida[1], Masahiro Yao[3]

1 Department of Urology, Kanagawa Cancer Center, Yokohama, Japan, 2 Departments of Urology and Renal Transplantation, Yokohama City University Medical Center, Yokohama, Japan, 3 Department of Urology, Yokohama City University Graduate School of Medicine, Yokohama, Japan

* kawahara@yokohama-cu.ac.jp

**Data Availability Statement:** The raw data underlying the findings of this study are included as a Supporting Information file.

## Abstract

### Background

Sarcopenia is defined as a low skeletal muscle volume. Recent studies have reported that sarcopenia is associated with a poor prognosis in various cancers. The purpose of this study is to evaluate the correlation between the psoas muscle volume and recurrence-free survival in patients with localized clear cell renal cell carcinoma (ccRCC).

### Methods

A total of 316 male patients with localized ccRCC who underwent radical nephrectomy at Yokohama City University Hospital (Yokohama, JAPAN) and Kanagawa Cancer Center (Yokohama, JAPAN) between 2002 and 2018 were enrolled in this study. The psoas muscle index (PMI) was calculated by normalizing the psoas muscle area on the contralateral side of the tumor on axial CT, which was calculated at the level of L4 ($mm^2$) divided by the square of the body height ($m^2$). We divided patients into two groups based on the median PMI ($409.64 mm^2/m^2$).

### Results

The lower PMI group showed poorer recurrence-free survival (RFS) than the higher PMI group (p = 0.030). Regarding 5-year RFS, a lower PMI was a significant predictor of recurrence (p = 0.022, hazard ratio (HR): 2.306) and a multivariate analysis revealed that a lower PMI (<median, p = 0.035, HR: 2.167), tumor size >4 cm (p = 0.044, HR: 2.341), and pathological stage >2 (p<0.001, HR: 3.660) were independent risk factors for poor RFS.

### Conclusions

The presence of sarcopenia (lower PMI) was found to be associated with poor RFS in male ccRCC patients. The PMI might serve as a measure of patient frailty and might be useful for prognostic risk stratification in ccRCC.

**Funding:** The author(s) received no specific funding for this work.

**Competing interests:** The authors have declared that no competing interests exist.

**Abbreviations:** ccRCC, clear cell RCC; CSS, cancer-specific survival; CT, computed tomography; OS, Overall survival; PMI, Psoas muscle index; PMV, psoas muscle volume; RCC, renal cell carcinoma; RFS, recurrence-free survival; SSIGN score, Stage Size Grade Necrosis score; UISS, University of California Integrated Staging System.

## Introduction

Renal cell carcinoma (RCC) is the most common malignant tumor of the adult kidney, accounting for 3.8% of all new cancers [1]. Resection of localized RCC is recommended as the only treatment for a complete cure. However, up to 30% of patients undergoing curative surgery will develop metastatic RCC [2, 3]. Surveillance can identify local recurrence or metastasis at an early stage. In patients with metastatic disease, extended tumor growth can limit the chance for surgical resection, considered the standard therapy in cases of resectable—and preferably solitary—lesions. In addition, the early diagnosis of tumor recurrence may enhance the efficacy of systemic treatment if the tumor burden is low. In order to better predict the prognosis, various centers have developed integrative prognostic tools, such as the University of California Integrated Staging System (UISS) and the Mayo Clinic Stage Size Grade Necrosis (SSIGN) score [4, 5]. These tools are clinically useful and improve the assessment of prognosis.

Recent studies have identified sarcopenia (loss of skeletal muscle mass) as one of the important factors for recurrence-free survival (RFS) [6]. The psoas muscle index (PMI), which was a relatively simple method to represent skeletal muscle volume in whole body. This index is attracting attentions as one of the prognostic factors for malignancy, although it was popularized as an index to quantify sarcopenia at first. The skeletal muscle mass is under investigation as a recurrence risk factor for malignant melanoma, colorectal cancer, hepatocellular carcinoma and pancreatic cancer. [7–10]. However, whether sarcopenia is a risk factor for recurrence of RCC has remained unclear. There are several different musculature measurements that are used to quantify sarcopenia. The PMI, which is relatively simple to determine, is one of the measurements of sarcopenia.

This study examined the predictive value of the preoperative measurement of the PMI for localized clear cell RCC (ccRCC) for predicting the RFS after curative nephrectomy.

## Materials and methods

We retrospectively assessed the importance of the PMI as a prognostic factor in male ccRCC patients. The primary endpoint was the RFS, the exposure variable was the PMI, and the overall survival (OS) and cancer-specific survival (CSS) were secondary endpoints.

### Patients

We examined consecutive male ccRCC patients who were treated with radical or partial nephrectomy with curative intent at Kanagawa Cancer Center (Yokohama, JAPAN) and Yokohama City University Hospital (Yokohama, JAPAN) from August 2002 to February 2018. Female patients were excluded from this study because female sex was associated with low rates of recurrence and cancer deaths and because the PMI is lower in women than in men, according to our previous reports [11]. In our previous study, the PMI was lower in women than in men (p<0.001) [S1 Fig] therefore, to exclude gender differences, we excluded women. In our cohort, only 9 of the 75 women followed-up for more than 1 year showed recurrence (median: 56.4 months). Due to this low number of cases, there was no marked difference in PMI between the non-recurrence and recurrence groups (median: 288.5 vs. 286.6, p = 0.463). Clinical and pathological data were collected, including the tumor characteristics, postoperative RFS, OS and CSS rates. All cases were staged preoperatively with chest and abdominal CT. A bone scan and brain imaging were performed when indicated by symptoms. The pathological stage was reassigned according to the 2009 TNM staging system. Patients with hereditary RCC were excluded.

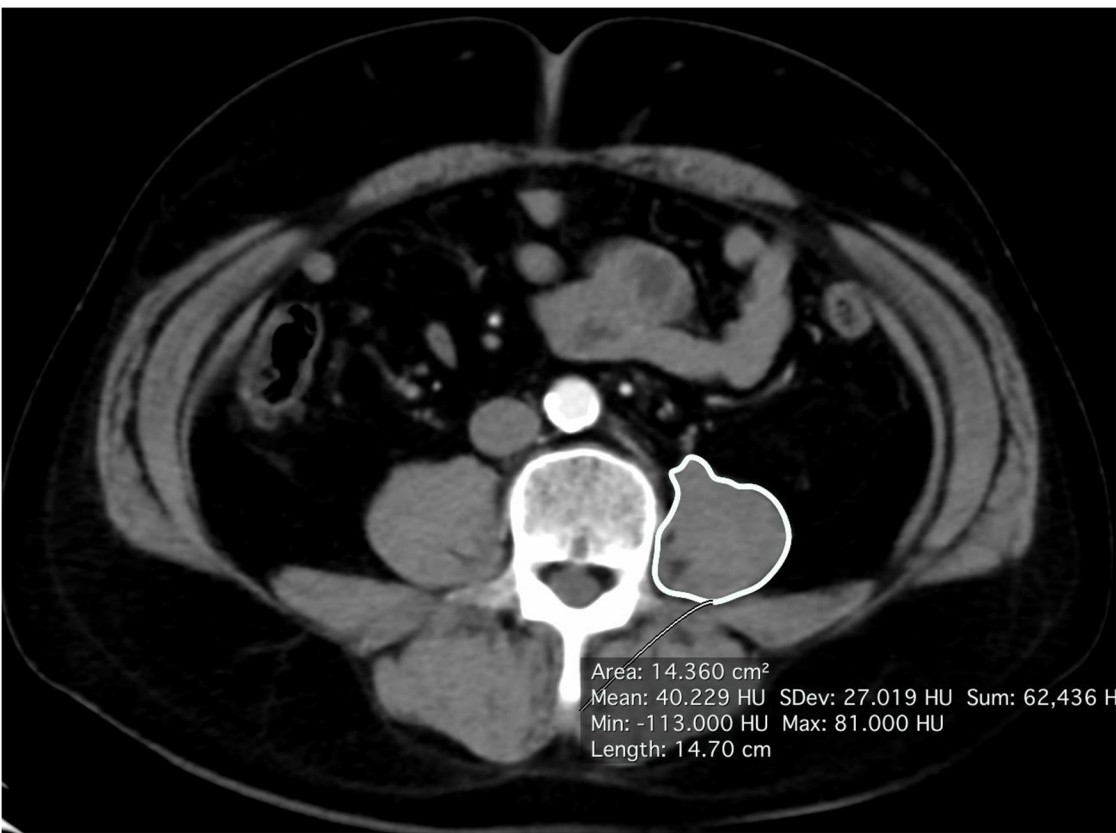

Area: 14.360 cm²
Mean: 40.229 HU  SDev: 27.019 HU  Sum: 62,436 H
Min: -113.000 HU  Max: 81.000 HU
Length: 14.70 cm

**Fig 1. Psoas muscle area calculated by axial CT.**

### Assessment of the PMI and psoas muscle volume (PMV)

We measured the cross-sectional areas of the major psoas muscle in contralateral side of the tumor at the level of L4 by manual tracing of a preoperative CT scan (Fig 1), and then PMI ($mm^2/m^2$) was calculated by normalizing the psoas muscle area ($mm^2$) divided by the square of the body height ($m^2$). The PMV ($cm^3$) on the contralateral side of the tumor on a preoperative CT scan was measured using the Osirix software program (Pixmeo, Geneva, Switzerland) (Fig 2). To assess the correlation between the PMV and PMI, we sampled 11 patients; their background characteristics are shown in S1 Table.

### Follow-up

Patients were generally followed every 3 to 6 months for the first 2 years after primary renal surgery, every 6 months from the second through fifth year, and annually thereafter. Disease recurrence was defined as radiographic evidence of disease on CT, magnetic resonance imaging, or a bone scan. Equivocal imaging findings were followed by biopsy of the suspected lesion and were classified as disease recurrence after pathological confirmation.

### Statistical analyses

The RFS, OS and CSS periods were calculated from the date of nephrectomy to the date at recurrence or last follow-up. The recurrence rates were estimated using the Kaplan–Meier method, and the resultant curves were statistically tested by the log-rank method. A Cox proportional hazards model was used for the univariate and multivariate analyses. The correlation

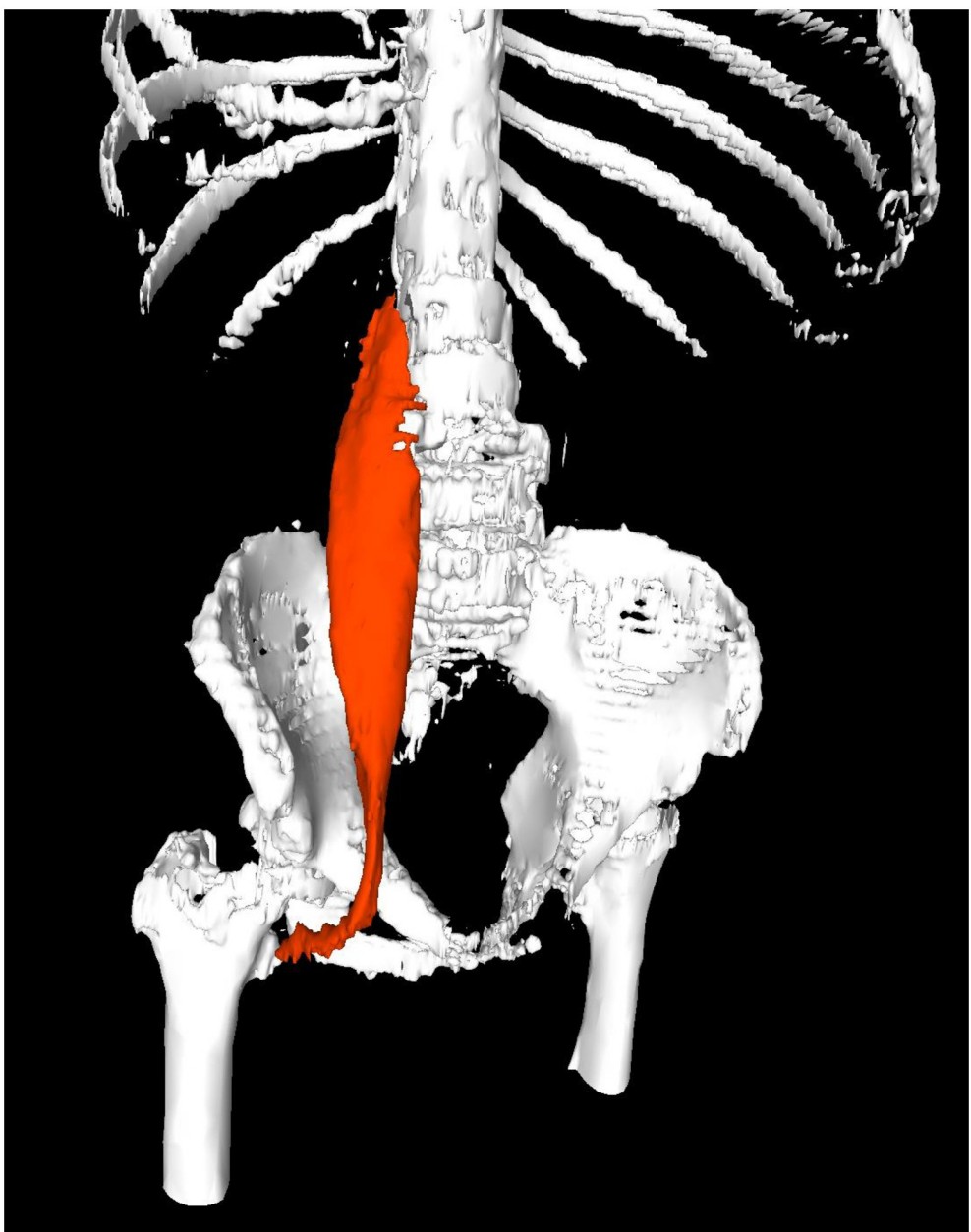

**Fig 2. Psoas muscle volume.**

between the PMV and PMI was assessed using Spearman's correlation coefficient analysis. P values of $< 0.05$ were considered to indicate statistical significance in all statistical tests. The statistical analyses were performed using the SPSS (version 25.0 SPSS Inc., Chicago, IL, USA) and GraphPad Prism (La Jolla, CA, USA) software programs.

## Ethical statement

This study was carried out in compliance with the Declaration of Helsinki and was approved by the Institutional Review Board of Yokohama City University Hospital (B160101010). This study is retrospective observational study and IRB did not require written patients consent.

**Table 1. Patient and tumor characteristics at primary radical surgery.**

| Variables | | Number (%) or median (range) |
|---|---|---|
| Numbers of patients | | 316 |
| Follow up periods (months) | | 68.8 (0–182) |
| Age at diagnosis (years) | | 63 (22–88) |
| Surgical procedure | Nephrectomy | 202 (63.9%) |
| | Partial nephrectomy | 114 (36.1%) |
| Affected Kidney | Right | 168 (53.2%) |
| | Left | 148 (46.8%) |
| ECOG PS | 0 | 292 (93.9%) |
| | ≥1 | 19 (6.1%) |
| Pathological Stage* | I | 246 (77.8%) |
| | II | 9 (2.8%) |
| | III | 61 (19.3%) |
| | IV | 0 (0.0%) |
| Tumor size | ≤4cm | 216 (68.4%) |
| | >4cm | 100 (31.6%) |
| Nuclear grade | G1 | 57 (18.0%) |
| | G2 | 179 (56.6%) |
| | G3 | 66 (20.9%) |
| | G4 | 14 (4.4%) |

* Pathologic stage: according to the 2009 TNM staging system

# Results

## Patient characteristics

A total of 316 patients were enrolled in this study. The median postoperative follow-up period was 68.8 months. Table 1 showed the clinicopathologic features of the enrolled patients. In this cohort, one patient received adjuvant treatment with pazopanib. A total of 49 patients (15.5%) had disease recurrence, 38 patients (11.7%) died of any cause, and there were 15 (4.7%) cancer deaths within the observation period. The 5-year RFS rate was 86.4% and the 5-year OS and CSS rates were 90.8% and 96%, respectively.

## The preoperative PMI predicted the RFS within 5 years after primary surgery

A small number of patients were analyzed to confirm the correlation between the PMI and the PMV. There was a strong correlation between the PMI and PMV ($R^2$ = 0.9502) (Fig 3). We therefore used the PMI instead of the PMV, because the method was easy to apply and because it could be determined without any additional software programs.

The PMI in the recurrence group was significantly lower than that in the no-recurrence group (p = 0.038) (Fig 4). The baseline characteristics of the recurrence and non-recurrence groups are shown in S2 Table. When we divided the patients into two groups according to the median PMI (409.64), the rate of recurrence in the lower PMI group was significantly higher than that in the higher PMI group (p = 0.030, HR 1.905 [95% CI: 1.065–3.408]). The OS and CSS rates in the lower and higher PMI groups did not differ to a statistically significant extent (p = 0.482 and p = 0.367, respectively). Table 2 showed the patient characteristics of the low and high PMI groups. When the results were analyzed focusing on recurrence within 5 years

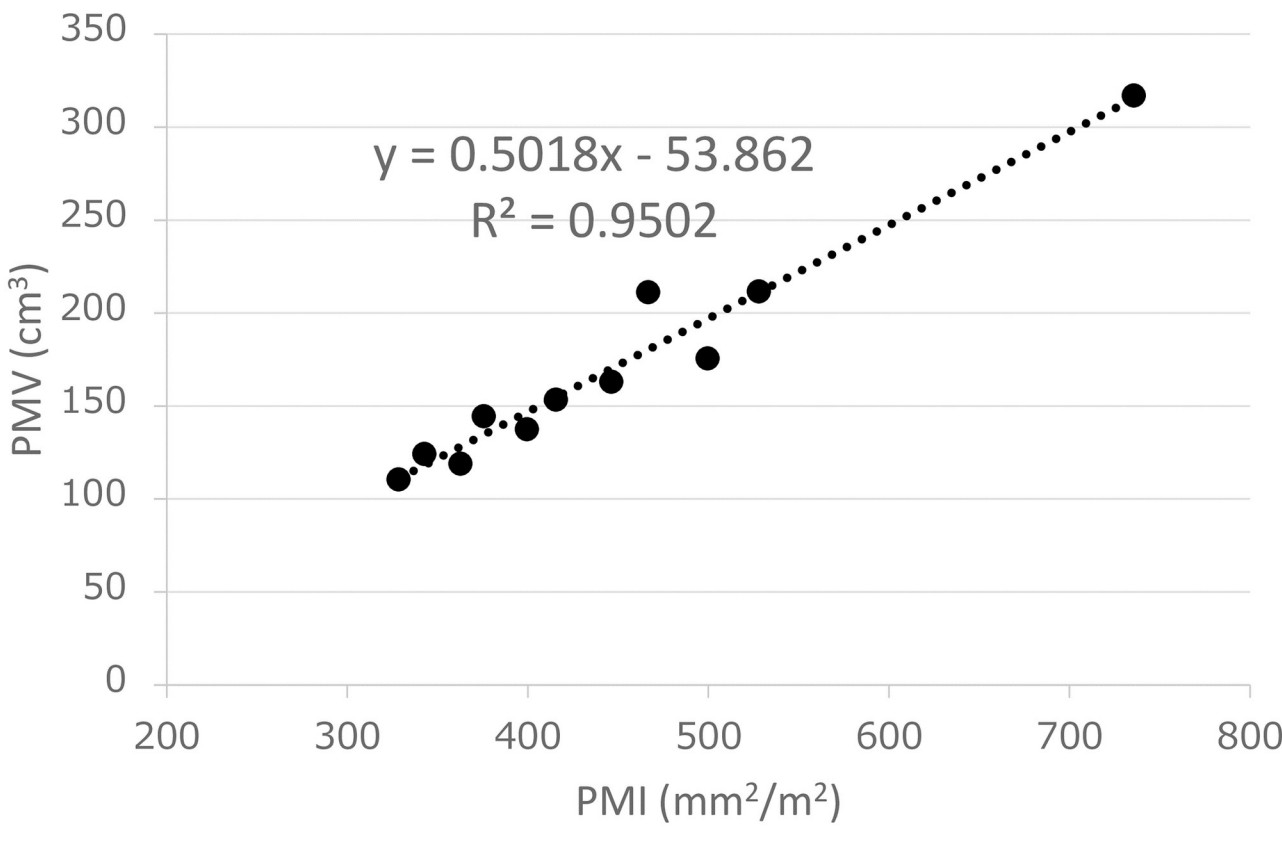

**Fig 3. The correlation between PMI and PMV.**

after surgery, a lower PMI value was a significant predictor of recurrence (P = 0.022, HR 2.306 (95% CI: 1.129–4.708)) (Fig 5). A multivariate analysis revealed that high stage, larger tumor size and lower PMI were independent risk factors for poor RFS (Table 3 and S3 Table).

## The PMI at 5 years after surgery

We analyzed 146 patients without recurrence at 5 years after surgery and measured the PMI again at 5 years after surgery (median 60.1 months after surgery, range: 52.5–66.0 months). When the patients were divided into two groups according to the PMI using the same cutoff value (409.64), there was a tendency toward recurrence in the low PMI group (p = 0.077, HR 2.686 [95% CI: 0.898–8.035]) (Fig 6).

Regarding the PMI values at the operation and at 5 years after surgery, there were 45 patients (30.8%) in whom the PMI increased by ≥5%, 58 patients (39.7%) whose values changed by -5% to 5%, and 43 patients (29.5%) whose values decreased by ≥5%. Two (4.4%), 6 (10.3%) and 6 (14.0%) patients of the ≥5% increase, -5% to 5% change, and ≥5% decrease groups had recurrence at 5 or more years after surgery, respectively (not significant).

## Discussion

In the present study, we assessed the prognostic value of the preoperative PMI in patients with localized RCC who underwent partial or radical nephrectomy and demonstrated that a lower PMI was significantly associated with shorter RFS. The results demonstrated that a lower PMI is an independent risk factor for recurrence in RCC patients after nephrectomy. To the best of

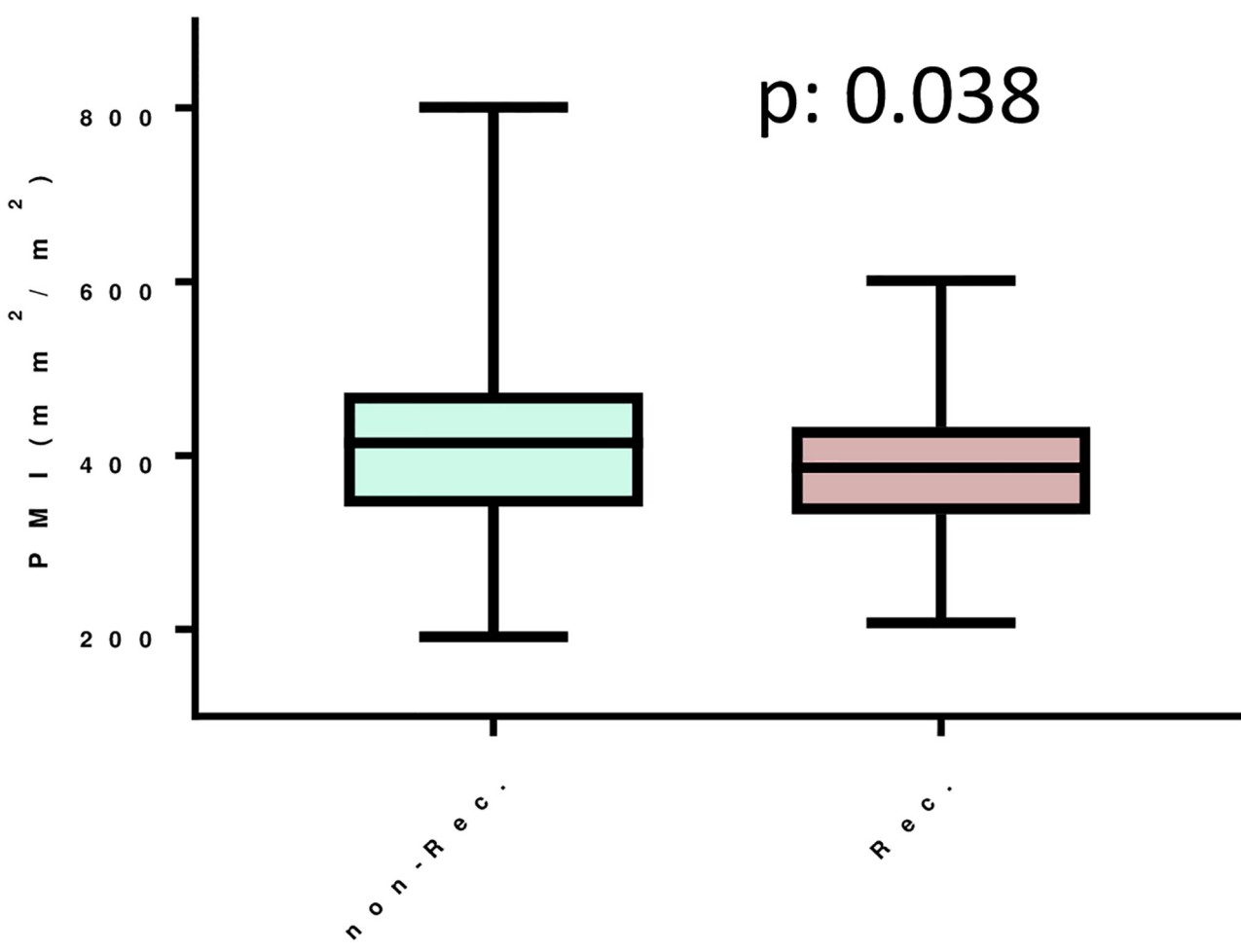

**Fig 4. PMI in non-recurrent and recurrent patients.**

**Table 2. Psoas muscle index and parameters.**

| | | Low PMI (n = 158) | High PMI (n = 158) | p value |
|---|---|---|---|---|
| | | Number (%) | | |
| Age | ≤60yrs. | 41 (25.9%) | 84 (53.2%) | <0.001 |
| | >60yrs. | 117 (74.1%) | 74 (46.8%) | |
| Site | Right | 83 (52.5%) | 85 (53.8%) | 0.822 |
| | Left | 75 (47.5%) | 73 (46.2%) | |
| PS | 0 | 143 (92.3%) | 149 (95.5%) | 0.231 |
| | ≥1 | 12 (7.7%) | 7 (4.5%) | |
| Stage | 1, 2 | 123 (77.8%) | 132 (83.5%) | 0.200 |
| | 3 | 35 (22.2%) | 26 (16.5%) | |
| Size | ≤4cm | 103 (65.2%) | 113 (71.5%) | 0.226 |
| | >4cm | 55 (34.8%) | 45 (28.5%) | |
| Grade | 1, 2 | 111 (70.3%) | 125 (79.1%) | 0.070 |
| | 3, 4 | 47 (29.7%) | 33 (20.9%) | |

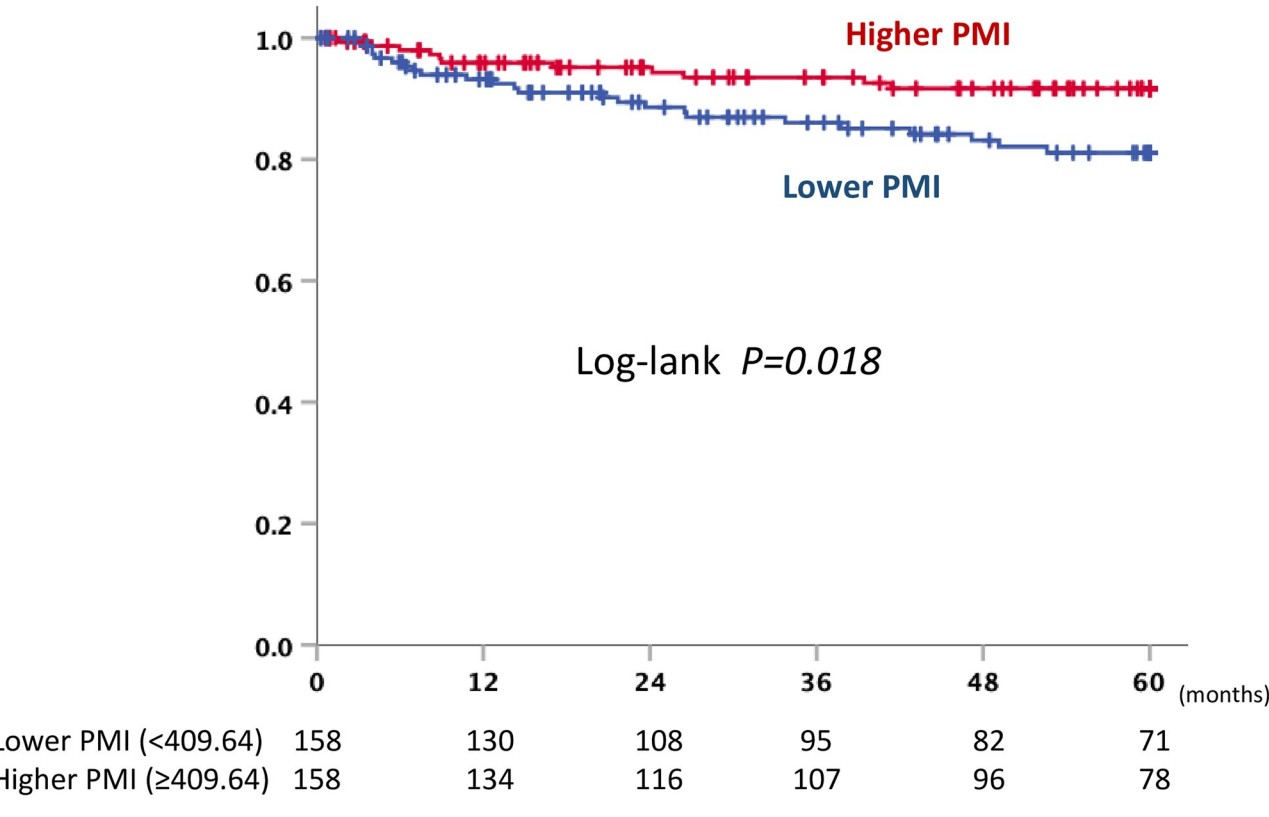

**Fig 5. Recurrence-free survival curve within 5 years after nephrectomy in higher and lower PMI.**

our knowledge, this is the first study to investigate the impact of PMI on recurrence in patients with RCC.

The PMI, which can be measured relatively simply, is one of the indices used to assess sarcopenia. A low PMI is regarded as a proxy for low muscle volume and our results showed a

**Table 3. Recurrence risk factor within 5 years after nephrectomy.**

|  |  | N | Univariate | | Multivariate | |
|---|---|---|---|---|---|---|
|  |  |  | HR (95%CI) | p value | HR (95%CI) | p value |
| Age | ≤60yrs. | 125 | 1 | 0.113 |  |  |
|  | >60yrs. | 191 | 1.811 (0.869–3.773) |  |  |  |
| Site | Right | 168 | 1 | 0.907 |  |  |
|  | Left | 148 | 1.040 (0.535–2.023) |  |  |  |
| PS | 0 | 292 | 1 | 0.054 |  |  |
|  | ≥1 | 19 | 2.797 (0.980–7.979) |  |  |  |
| Stage | 1, 2 | 255 | 1 | <0.001 | 1 | <0.001 |
|  | 3 | 61 | 6.775 (3.466–13.243) |  | 3.660 (1.671–8.015) |  |
| Size | ≤4cm | 216 | 1 | <0.001 | 1 | 0.044 |
|  | >4cm | 100 | 4.452 (2.289–8.660) |  | 2.341 (1.024–5.353) |  |
| Grade | 1, 2 | 236 | 1 | <0.001 | 1 | 0.172 |
|  | 3, 4 | 80 | 3.768 (1.940–7.318) |  | 1.660 (0.802–3.437) |  |
| PMI | High | 158 | 1 | 0.022 | 1 | 0.035 |
|  | Low | 158 | 2.306 (1.129–4.708) |  | 2.167 (1.056–4.444) |  |

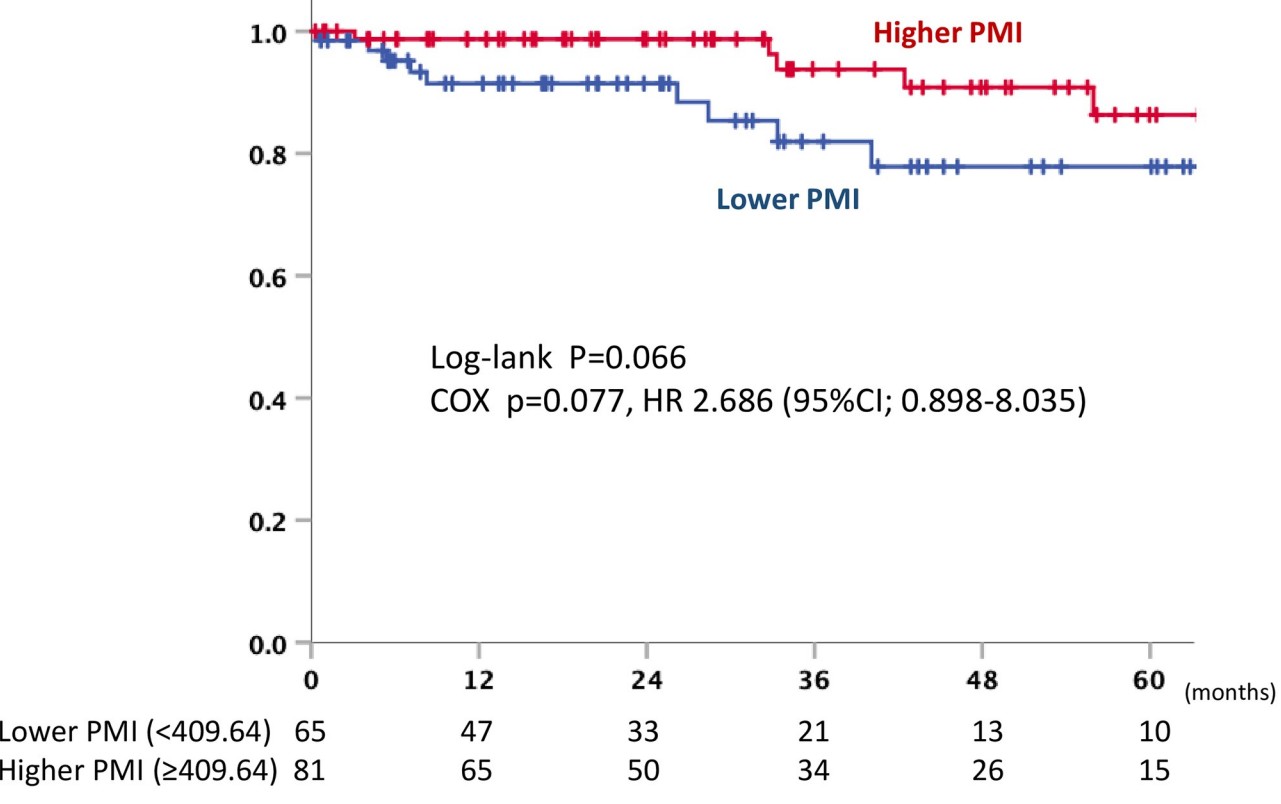

**Fig 6. Overall survival curve within 5 years after nephrectomy in higher and lower PMI.**

good correlation between the PMI and PMV. Sarcopenia is defined as a low skeletal muscle volume. Sarcopenia can be considered 'primary' (or age-related sarcopenia) when no other cause is evident other than ageing itself, or 'secondary' when one or more causes (*e.g.*, activity-related, disease-related and nutrition-related) are evident [12]. Many studies in recent years have shown an association between sarcopenia and a poor prognosis in various cancers [6]. Sarcopenia is closely related to patient frailty and has been reported to be associated with immune functions such as immunosenescence [13].

The UISS and SSIGN scores are two widely validated prognostic scoring models for RCC. The UISS score, which was proposed by the University of California, Los Angeles, is determined according to the tumor stage, Fuhrman grade and ECOG-PS [4]. The SSIGN score, which was proposed by the Mayo Clinic, is determined based on the T stage, nodal disease, tumor size, nuclear grade, presence or absence of tumor necrosis, and the presence or absence of metastasis [5]. We investigated the PMI as a recurrence factor in comparison to these major prognostic factors. The present study suggested that a lower PMI was an independent risk factor for ccRCC after curative surgery. In clear cell renal cancer, cytokine therapy is still a recommended treatment for favorable-risk cancer with lung metastasis. Our findings suggest that a patient's underlying vitality (or lack thereof) is an important risk factor for recurrence in RCC. It was reported that a shift in balance between the tumor and host was important for disease progression of melanoma [7]. These results may suggest that objective measures of frailty, such as sarcopenia, may be a better method for evaluating the physiological host reserve and overall wellness.

Only one study has investigated the correlation between sarcopenia and the prognosis of localized RCC [14]. They investigated the correlation between the skeletal muscle index and the prognosis of localized RCC and reported that sarcopenia was independently associated with OS and CSS after radical nephrectomy for renal cell carcinoma. However, it was not associated with PFS. We considered that the fact that our study population was restricted to male patients was the reason for the discrepancy in the outcome. In female patients, changes in hormonal balance, such as menopause, are thought to cause changes in muscle mass, and may not simply reflect the frailty of the patient. As our study population only included men, the PMI strongly reflected patient frailty. Thus, this might have been responsible for the difference in PFS. In contrast, the small number of events, was likely the reason why no difference between OS and PMI was observed in our study.

We further analyzed the patients who had no recurrence for 5 years an additional PMI measurement. Although the result was not statistically significant, the PMI tended to be correlated with subsequent recurrence. The results suggested that the PMI may be a biomarker for predicting recurrence, and further studies, with longer follow-up periods are warranted to investigate the correlation between the PMI and recurrence over time.

The present study was associated with some limitations. It was a non-randomized retrospective study and enrolled a relatively small number of patients. We analyzed the PMI as an indicator of sarcopenia and the PMI was associated with the PMV; however, we did not evaluate whether or not the PMI was correlated with the total skeletal muscle volume. In RCC, the psoas muscle area may not be accurately evaluated due to the influence of the tumor; thus, we only evaluated the area of the major psoas muscle on the contralateral side of the tumor. The PMI cutoff value is dependent on race and the exact cutoff point for each race is still unknown. However, the current study revealed an important finding: a low PMI was an independent predictor for recurrence in male RCC patients. Furthermore, our method is easy to perform, and no additional procedures for measuring the psoas muscle volume are required for most patients undergoing radical nephrectomy. The present study indicated that the PMI, a simple and easily measurable index of sarcopenia, was strongly correlated with short-term recurrence. Due to the small number of patients in the cohort and the retrospective, observational nature of the study, a further prospective study is needed in order to confirm these results.

In conclusion, the present study suggested a significant correlation between the preoperative PMI and postoperative disease recurrence. Thus, the PMI, which is easy to measure on preoperative abdominal CT scans, may improve the accuracy of the predicted prognosis after radical nephrectomy. In ccRCC, in addition to the tumor biology the host biology is an important factor in disease progression. The PMI might serve as a measure of patient frailty and may therefore be useful for prognostic risk stratification.

## Supporting information

**S1 Fig. The differences in the PMI between male and females.**
(PPTX)

**S1 Table.**
(DOCX)

**S2 Table.**
(DOCX)

**S3 Table.**
(DOCX)

**S1 Data.**
(PDF)

## Acknowledgments

We would like to thank Editage and Japan Medical Communication for performing the
English language editing of this manuscript.

## Author Contributions

**Conceptualization:** Takashi Kawahara, Susumu Umemoto, Masahiro Yao.

**Data curation:** Go Noguchi, Kota Kobayashi, Sohgo Tsutsumi, Shinji Ohtake, Kimito Osaka.

**Investigation:** Go Noguchi, Takashi Kawahara, Susumu Umemoto, Noboru Nakaigawa.

**Supervision:** Hiroji Uemura, Takeshi Kishida, Masahiro Yao.

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
