## [Decision Letter · Decision Letter 0]

3 Oct 2019

PONE-D-19-21225

A lower psoas muscle volume was associated with a higher rate of recurrence in male clear cell renal cell carcinoma

PLOS ONE

Dear Dr. Kawahara,

Thank you for submitting your manuscript to PLOS ONE. After careful consideration, we feel that it has merit but does not fully meet PLOS ONE’s publication criteria as it currently stands. Therefore, we invite you to submit a revised version of the manuscript that addresses the points raised during the review process.

Specifically, the analytical aspects of the study raised by the reviewers (especially Reviewer #2) must be addressed.

We would appreciate receiving your revised manuscript by Nov 17 2019 11:59PM. To enhance the reproducibility of your results, we recommend that if applicable you deposit your laboratory protocols in protocols.io, where a protocol can be assigned its own identifier (DOI) such that it can be cited independently in the future. For instructions see: http://journals.plos.org/plosone/s/submission-guidelines#loc-laboratory-protocols

We look forward to receiving your revised manuscript.

Kind regards,

Biniam Kidane

Academic Editor

PLOS ONE

Journal Requirements:

1. Thank you for including your ethics statement:  "This study was carried out in compliance with the Declaration of Helsinki and was approved by the Institutional Review Board of each institution (B160101010)."

2. Please provide additional details regarding participant consent. In the ethics statement in the Methods and online submission information, please ensure that you have specified

(a) whether consent was informed and

(b) what type you obtained (for instance, written or verbal). If your study included minors, state whether you obtained consent from parents or guardians. If the need for consent was waived by the ethics committee, please include this information.

(c) Once you have amended this/these statement(s) in the Methods section of the manuscript, please add the same text to the “Ethics Statement” field of the submission form (via “Edit Submission”).

3. We noticed minor instances of text overlap with the following previous publication(s), which need to be addressed:

https://link.springer.com/article/10.1007/s10585-018-9883-0

http://www.oncotarget.com/index.php?journal=oncotarget&op=view&page=article&path%5B%5D=22688&path%5B%5D=71664

In your revision please ensure you cite all your sources (including your own works), and quote or rephrase any duplicated text outside the methods section. Further consideration is dependent on these concerns being addressed.

Reviewers' comments:

Reviewer's Responses to Questions

**Comments to the Author**

1. Is the manuscript technically sound, and do the data support the conclusions?

Reviewer #1: Yes

Reviewer #2: Yes

2. Has the statistical analysis been performed appropriately and rigorously? 

Reviewer #1: Yes

Reviewer #2: Yes

3. Have the authors made all data underlying the findings in their manuscript fully available?

Reviewer #1: No

Reviewer #2: Yes

4. Is the manuscript presented in an intelligible fashion and written in standard English?

Reviewer #1: Yes

Reviewer #2: Yes

5. Review Comments to the Author

Reviewer #1: I commend the authors on attempting to elucidate the prognostic value of sarcopenia (with a psoas muscle index) in male patients with clear cell renal cell cancer. This is a retrospective cohort study of 316 patients undergoing curative intent surgery for ccRCC. I have appended a few minor suggestions for improving this paper prior to submission below:

Methods

1. Primary outcome and exposure variable needs to be clearly stated in the methods section as per the STROBE statement for reporting observational studies

2. Method of determining correlation between PMV and PMI, including number of patients in this sub-cohort, requires further description in the methods section

3. The authors should attempt to compare the baseline clinical and oncologic metrics of patients with and without recurrent disease. The average PMI appears to be lower in patients with a recurrence, however, it is important to first determine whether this is a result of more advanced disease in this group.

Discussion

1. The authors must be cautious not be overstate their findings. The findings of this retrospective cohort study, with a limited sample size of patients with a clinical recurrence, suggests a correlation between PMI and RFS and is good first step in determining the prognostic value of sarcopenia in RCC. These findings however, require prospective validation.

Reviewer #2: ONE-D-19-21225

This is a retrospective cohort study of men with localized Clear Cell RCC undergoing surgical resection wherein the authors sought to determine the association between sarcopenia and recurrence. The authors utilized psoas muscle index (PMI) as the exposure variable, and showed low PMI was associated with a higher risk of recurrence.

Methods:

1. It does not make sense to exclude Female patients because the rates of recurrence was low or because the psoas muscle index was low. What the was even rate in women? Analysis could be considered separately in women with a different cut off of PMI.

2. The authors need to include the following covariates in the multivariable model: Margin Status, sarcomatoid/rhabdoid histology, receipt of adjuvant therapy (e.g. sunitinib)

3. Please explain why the analysis was restricted to clear cell histology only.

6. PLOS authors have the option to publish the peer review history of their article (what does this mean?). If published, this will include your full peer review and any attached files.

Reviewer #1: Yes: Dr. Dhruvin Hirpara

Reviewer #2: No

---

## [Author Response · Author response to Decision Letter 0]

24 Oct 2019

Editorial office

PLoS One

Re: PONE-D-19-21225 "A lower psoas muscle volume was associated with a higher rate of recurrence in male clear cell renal cell carcinoma".

Dear Editor,

Thank you for your letter concerning the abovementioned manuscript. We are pleased to note the favorable comments of the reviewers and have revised the manuscript. Our point-by-point revisions are described below.

We would like to thank the Editor and reviewers again for their helpful comments and hope that the revised manuscript is acceptable for publication in PLoS One.

Respectfully yours,

Takashi Kawahara, M.D., Ph.D.

Department of Urology and Renal Transplantation

Yokohama City University Medical Center

4-57 Urafune, Minami-ku, Yokohama, Kanagawa, 2320024, Japan

Phone: +81-45-787-2679

Fax: +81-45-786-5656

E-mail: takashi_tk2001@yahoo.co.jp or kawahara@yokohama-cu.ac.jp

RESPONSE TO REVIEWERS

Reviewer 1

I commend the authors on attempting to elucidate the prognostic value of sarcopenia (with a psoas muscle index) in male patients with clear cell renal cell cancer. This is a retrospective cohort study of 316 patients undergoing curative intent surgery for ccRCC. I have appended a few minor suggestions for improving this paper prior to submission below:

Methods

1. Primary outcome and exposure variable needs to be clearly stated in the methods section as per the STROBE statement for reporting observational studies

Response: We have now added information about the exposure variable and primary outcome to the revised manuscript.

2. Method of determining correlation between PMV and PMI, including number of patients in this sub-cohort, requires further description in the methods section

Response: We have now added the patients’ background characteristics to the revised manuscript in the Methods section and added a supplementary table as follows:

3. The authors should attempt to compare the baseline clinical and oncologic metrics of patients with and without recurrent disease. The average PMI appears to be lower in patients with a recurrence, however, it is important to first determine whether this is a result of more advanced disease in this group.

Response: We have now added the following table to the revised manuscript:

 Recurrence (n=49) No recurrence (n=267) 

　 　 n (%) n (%) p

Age ≤60yrs. 13 (26.5%) 112 (41.9%) 

 >60yrs. 36 (73.5%) 155 (58.1%) 0.042 

Site Right 25 (51.0%) 143 (53.6%) 

 Left 24 (49.0%) 124 (46.4%) 0.744 

PS 0 40 (88.9%) 252 (94.7%) 

 ≥1 5 (11.1%) 14 (5.3%) 0.130 

Stage 1 & 2 23 (46.9%) 232 (86.9%) 

 3 26 (53.1%) 35 (13.1%) <0.001

Size ≤4cm 11 (22.4%) 176 (65.9%) 

 >4cm 38 (77.6%) 91 (34.1%) <0.001

Grade 1 & 2 24 (49.0%) 212 (79.4%) 

 3 & 4 25 (51.0%) 55 (20.6%) <0.001

PMI High 18 (36.7%) 140 (52.4%) 

　 Low 31 (63.3%) 127 (47.6%) 0.043 

Discussion

4. The authors must be cautious not be overstate their findings. The findings of this retrospective cohort study, with a limited sample size of patients with a clinical recurrence, suggests a correlation between PMI and RFS and is good first step in determining the prognostic value of sarcopenia in RCC. These findings however, require prospective validation.

Response: As you mention, this study is a retrospective observational study with a limited number of patients. We have now touched on these limitations in the revised manuscript.

Reviewer 2

This is a retrospective cohort study of men with localized Clear Cell RCC undergoing surgical resection wherein the authors sought to determine the association between sarcopenia and recurrence. The authors utilized psoas muscle index (PMI) as the exposure variable, and showed low PMI was associated with a higher risk of recurrence.

Methods:

1. It does not make sense to exclude Female patients because the rates of recurrence was low or because the psoas muscle index was low. What the was even rate in women? Analysis could be considered separately in women with a different cut off of PMI

Response: In our previous study, the PMI was lower in women than in men (p<0.001); therefore, to exclude gender differences, we excluded women. In our cohort, only 9 of the 75 women followed-up for more than 1 year showed recurrence (median: 56.4 months). Due to this low number of cases, there was no marked difference in PMI between the non-recurrence and recurrence groups (median: 288.5 vs. 286.6, p=0.463). We have now mentioned this in the revised manuscript and supplementary table 2.

2. The authors need to include the following covariates in the multivariable model: Margin Status, sarcomatoid/rhabdoid histology, receipt of adjuvant therapy (e.g. sunitinib)

Response: In the present cohort, only one case received adjuvant pazopanib treatment. Sarcomatoid was categorized as Grade 4. We therefore performed the multivariate analysis again and obtained the following results:

3. Please explain why the analysis was restricted to clear cell histology only.

Response: We initially evaluated this cohort in order to analyze cytokine efficacy. Cytokine therapy has been confirmed to be effective against melanoma and clear cell carcinoma but not non-clear cell carcinoma. The EAU guideline also recommends interferon therapy for favorable-risk clear cell renal cancer with lung metastasis but not non-clear cell renal cancer. We have now mentioned these points in the revised manuscript.

Journal Requirements:

1. Thank you for including your ethics statement: "This study was carried out in compliance with the Declaration of Helsinki and was approved by the Institutional Review Board of each institution (B160101010)."

2. Please provide additional details regarding participant consent. In the ethics statement in the Methods and online submission information, please ensure that you have specified

(a) whether consent was informed and

(b) what type you obtained (for instance, written or verbal). If your study included minors, state whether you obtained consent from parents or guardians. If the need for consent was waived by the ethics committee, please include this information.

(c) Once you have amended this/these statement(s) in the Methods section of the manuscript, please add the same text to the “Ethics Statement” field of the submission form (via “Edit Submission”).

3. We noticed minor instances of text overlap with the following previous publication(s), which need to be addressed:

In your revision please ensure you cite all your sources (including your own works), and quote or rephrase any duplicated text outside the methods section. Further consideration is dependent on these concerns being addressed.

Response: We have now ensured that the revised manuscript complies with the above.

---

## [Decision Letter · Decision Letter 1]

3 Dec 2019

A lower psoas muscle volume was associated with a higher rate of recurrence in male clear cell renal cell carcinoma

PONE-D-19-21225R1

Dear Dr. Kawahara,

We are pleased to inform you that your manuscript has been judged scientifically suitable for publication and will be formally accepted for publication once it complies with all outstanding technical requirements.

With kind regards,

Biniam Kidane

Academic Editor

PLOS ONE

Additional Editor Comments (optional):

Reviewers' comments:

Reviewer's Responses to Questions

**Comments to the Author**

1. If the authors have adequately addressed your comments raised in a previous round of review and you feel that this manuscript is now acceptable for publication, you may indicate that here to bypass the “Comments to the Author” section, enter your conflict of interest statement in the “Confidential to Editor” section, and submit your "Accept" recommendation.

Reviewer #1: All comments have been addressed

Reviewer #2: All comments have been addressed

2. Is the manuscript technically sound, and do the data support the conclusions?

Reviewer #1: Yes

Reviewer #2: Yes

3. Has the statistical analysis been performed appropriately and rigorously? 

Reviewer #1: Yes

Reviewer #2: Yes

4. Have the authors made all data underlying the findings in their manuscript fully available?

Reviewer #1: Yes

Reviewer #2: Yes

5. Is the manuscript presented in an intelligible fashion and written in standard English?

Reviewer #1: Yes

Reviewer #2: Yes

6. Review Comments to the Author

Reviewer #1: Thank you for your revisions. The manuscript has been appropriately amended and is acceptable for publication.

Reviewer #2: (No Response)

7. PLOS authors have the option to publish the peer review history of their article (what does this mean?). If published, this will include your full peer review and any attached files.

Reviewer #1: No

Reviewer #2: No

---

## [Editor Report · Acceptance letter]

10 Dec 2019

PONE-D-19-21225R1 

A lower psoas muscle volume was associated with a higher rate of recurrence in male clear cell renal cell carcinoma 

Dear Dr. Kawahara:

I am pleased to inform you that your manuscript has been deemed suitable for publication in PLOS ONE. Congratulations! Your manuscript is now with our production department. 

With kind regards,

on behalf of

Dr. Biniam Kidane 

Academic Editor

PLOS ONE